# DROSHA-Dependent miRNA and AIM2 Inflammasome Activation in Idiopathic Pulmonary Fibrosis

**DOI:** 10.3390/ijms21051668

**Published:** 2020-02-28

**Authors:** Soo Jung Cho, Mihye Lee, Heather W. Stout-Delgado, Jong-Seok Moon

**Affiliations:** 1Joan and Sanford I. Weill Department of Medicine, Weill Cornell Medical College, New York-Presbyterian Hospital, New York, NY 10011, USA; sjc9006@med.cornell.edu; 2Division of Pulmonary and Critical Care Medicine, Weill Cornell Medical College, New York, NY 10011, USA; 3Department of Integrated Biomedical Science, Soonchunhyang Institute of Medi-bio Science (SIMS), Soonchunhyang University, Chungcheongnam-do, Cheonan-si 31151, Korea; mihyelee@sch.ac.kr

**Keywords:** DROSHA, miRNA, AIM2 inflammasome, IPF

## Abstract

Idiopathic pulmonary fibrosis (IPF) is a chronic, progressive interstitial lung disease. Chronic lung inflammation is linked to the pathogenesis of IPF. DROSHA, a class 2 ribonuclease III enzyme, has an important role in the biogenesis of microRNA (miRNA). The function of miRNAs has been identified in the regulation of the target gene or protein related to inflammatory responses via degradation of mRNA or inhibition of translation. The absent-in-melanoma-2 (AIM2) inflammasome is critical for inflammatory responses against cytosolic double stranded DNA (dsDNA) from pathogen-associated molecular patterns (PAMPs) and self-DNA from danger-associated molecular patterns (DAMPs). The AIM2 inflammasome senses double strand DNA (dsDNA) and interacts with the adaptor apoptosis-associated speck-like protein containing a caspase recruitment domain (ASC), which recruits pro-caspase-1 and regulates the maturation and secretion of interleukin (IL)-1β and IL-18. A recent study showed that inflammasome activation contributes to lung inflammation and fibrogenesis during IPF. In the current review, we discuss recent advances in our understanding of the DROSHA–miRNA–AIM2 inflammasome axis in the pathogenesis of IPF.

## 1. Introduction

Idiopathic pulmonary fibrosis (IPF) is a chronic and progressive lung disease that is the most common interstitial lung disease. IPF is a lethal disorder of unknown etiology and has an incidence in northern America estimated between six and 18 per 100,000 inhabitants [1]. Since the time of diagnosis, patients with IPF have a 2–3 years median survival period [2]. IPF has been linked to certain areas of chronic inflammation, abnormal immune responses, cell injury, cell proliferation, and impaired wound healing, which leads to respiratory failure and death [3,4]. The current approach to IPF diagnosis is multidimensional including clinical, radiological, and histopathological features. Using the current guidelines, a confident diagnosis of IPF can be made based on proper clinical history and subpleural, basal predominant reticular opacities with traction bronchiectasis on high resolution computed tomography [5]. For the patients with non-diagnostic clinical and radiographical findings, surgical lung biopsy is recommended and often shows spatial heterogeneity with areas of subpleural and paraseptal fibrosis and honeycombing [6]. Although the etiology of IPF remains unknown, there are several potential risk factors such as smoking, various types of infection, and exposure of environmental particles. Additionally, genetic alteration by specific genes including surfactant protein A2, surfactant protein C, ELMO/CED-12 domain containing 2, mucin 5b, and telomerase genes was associated with IPF [7,8]. In this context, the role of microRNA (miRNA), a family of small non-coding RNAs, is important for the regulation of genes through post-transcriptional modification, which causes various cellular changes such as cell differentiation, cell proliferation, and the interaction between cells and cells [9]. More recently drosha ribonuclease III (DROSHA) that processes miRNA, has been recognized as a key molecule in the pathogenesis of IPF [10]. In parallel with this emerging interest in the molecular mechanisms that generate miRNA, there has been consistent interest in controlling inflammation including AIM2 inflammasome activation in chronic inflammatory conditions such as IPF. In this review, we discuss recent advances in our understanding of the DROSHA–miRNA–AIM2 inflammasome axis in the pathogenesis of IPF.

## 2. DROSHA-Dependent miRNA Biogenesis in Idiopathic Pulmonary Fibrosis (IPF)

### 2.1. The Role of DROSHA in IPF

DROSHA is a nuclease of the RNase III family [11,12]. As a core nuclease in nucleus, DROSHA plays a role in the initiation of miRNA maturation (Figure 1). DROSHA produces precursor miRNA (pre-RNA) via cleavage of primary miRNA (pri-miRNA) transcripts that are transported to cytoplasm and affect the regulation of molecular targets [13]. The DROSHA protein is localized into the nucleus by phosphorylation and leads to ultimate miRNA processing function [14]. Treatment with MG-132, a selective proteasome inhibitor, also increases the level of DROSHA, which suggests that DROSHA is regulated by post-translational modification [15]. DGCR8 (DiGeorge syndrome critical region 8), a cofactor of “Microprocessor” was reported to stabilize DROSHA through protein-protein interactions [16]. Under stress conditions, p38MAPK phosphorylates DROSHA, which decreases the interaction of DROSHA with DGCR8 [17]. The next critical step in miRNA biogenesis is pre-miRNA processing by DROSHA, which generates pre-miRNA that defines the miRNA sequences embedded in long pre-miRNAs [18,19]. Recently, we have reported for the first time that the protein expression of DROSHA was increased in alveolar macrophages of patients with IPF and mice with bleomycin-induced pulmonary fibrosis [10]. Conversely, DGCR8, a cofactor of DROSHA, protein expression was comparable between IPF and the control lungs [10]. After pre-miRNAs are exported to cytoplasm by exportin 5 in a Ran-GTP-dependent manner [20], Dicer, which is a well conserved RNAse III endoribonuclease, performs the further process in cytoplasm as the maturation of miRNAs makes progress [21]. Interestingly, Dicer was noted to be downregulated in fibroblasts from IPF lung [22]. This suggests that DROSHA, DGCR8, and Dicer play independent roles in IPF pathogenesis. 

### 2.2. The Role of miRNA in IPF

DROSHA is involved in maturation or processing and not the biogenesis of microRNAs (miRNAs), small non-coding RNA molecules that have 18–25 nucleotides [9,23]. The miRNAs are expressed in the various types of organs and cells [18,19,24]. The miRNAs could change the inflammatory responses via the regulation of gene expression by post-transcriptional modifications that cause degradation and inhibition of the translation of the molecular target [18,19,24]. Recent studies have shown that mature miRNAs could be potential biomarkers for diagnosis in many human diseases such as cancers, cardiovascular disease, and neurodegenerative disease [25,26,27,28]. In a study of IPF, the changes of miRNAs were demonstrated in the analysis of miRNA arrays containing probes for 450 miRNAs with isolated RNA from the lungs from patients with IPF or the control [29]. It is reported that approximately 10% of miRNAs are significantly changed in human IPF lungs (Figure 2). Pandit et al. showed that downregulation of let-7d in IPF and inhibition of let-7d caused increases in profibrotic effects including the production of N-cadherin-2, vimentin, α-SMA, and HMGA1. In addition, Eliot et al. reported that let-7 downregulates ligand-independent estrogen receptor (ER)-mediated pulmonary fibrosis and this study suggested that let-7 dependent ER expression has an important role in male-predominant fibrotic lung disease [30]. Additionally, in a murine fibrosis model, a relative abundance of miRNA levels in bleomycin-treated lung was demonstrated [31]. Wei et al. showed that TGF-β1-induced miR-133a inhibits myofibroblast differentiation and pulmonary fibrosis by downregulating a-SMA, CTGF, and collagens [32]. The targets of miR-133a are TGF-β receptor 1, CTGF, and collagen type 1-a1. Bahudhanapati et al. reported that miR-144-3p is upregulated in IPF fibroblasts compared with the control fibroblasts [33]. miR-144-3p controls the expression of relaxin/insulin-like family peptide receptor 1(RXFP1) in fibroblasts from lungs with IPF [34]. Recent studies have identified the interaction between long noncoding RNAs (lncRNAs) and miRNAs. Jiang et al. showed that inhibition of pulmonary fibrosis-regulatory lncRNA (PFRL) prevents pulmonary fibrosis by disrupting the miR-26a/smad2 loop [34]. In addition, another lncRNA, pulmonary fibrosis-associated lncRNA (PFAL) was found to be upregulated during lung fibrosis in vitro and in vivo fibrosis models [35]. Mechanistically, PFRL promoted lung fibroblast activation by acting as a competing endogenous RNA for miR-18a. Liu et al. analyzed the function and regulatory mechanism of miR-708-3p and showed that miR-708-3p is decreased in IPF lungs [36]. They reported that miR-708-3p directly controls a disintegrin and metalloproteinase 17 (ADAM17) [36]. IPF is the most common interstitial lung fibrosis that shares similar pathways with other fibrotic disease such as scleroderma [37]. Several microRNAs including, but not limited to miR-29, miR-21-5p, and miR-26a-5p play important roles in lung fibrosis and disease progression [38,39,40,41,42,43].

What are the targets of these miRNAs in the pathogenesis of lung fibrosis? Lung epithelial cells and fibroblasts play a critical role in the development and progression of lung fibrosis [44]. Let-7d, miR-29b, miR-26a, and miR-98 mediates antifibrotic effect by regulating myofibroblast activation and differentiation [29,42,45,46]. Conversely, miR-21, miR-424, and miR-145 are pro-fibrotic by suppressing negative regulators of TGF-β signaling [47,48,49]. Another important pathway that plays a central role in IPF development are aberrant inflammatory responses. There are several miRNAs that have been related to pulmonary fibrosis, out of which only miR-29 has been shown to regulate the innate immune response in vivo [45]. The administration of miR-29 attenuates both inflammatory response and fibrosis in the bleomycin induced lung fibrosis model. Currently, miRNAs have a critical role as key gene regulators that control fibrosis and inflammation, which in turn are potential targets for IPF disease treatment (Figure 2).

## 3. The Activation of Inflammasomes in IPF

### 3.1. The Role of Inflammasome-Dependent Inflammation in IPF

Alveolar macrophages are resident innate immune cells in the airways [50,51,52,53]. Alveolar macrophages play a key role in antimicrobial phagocytosis and in the pathogenesis of fibrotic lung disease as a critical type of cell in the lung [50,51,52,53]. Inflammasomes are multiprotein complexes, expressed in innate immune cells such as macrophages. Inflammasomes make inflammatory responses by the recognition of pathogen-associated molecular patterns (PAMPs), which come from bacterial, virus, and fungi, and danger-associated molecular patterns (DAMPs) that come from injured or dead cells [54,55]. The types of inflammasomes are distinguished by the combination of specific sensor proteins including NOD-, LRR-, and pyrin domain-containing 1 (NLRP1), Nod-like receptor protein 3 (NLRP3), NLRC4, or PYHIN (pyrin and HIN domain-containing protein) family members absent in melanoma 2 (AIM2), with the adaptor protein, adaptor apoptosis-associated speck-like protein containing a caspase recruitment domain (ASC), and the effector protein, capasae-1 [56,57,58]. The activation of NLRP3 inflammasome by the inflammasome complex formation is induced by various bacteria and viral particles, whereas AIM2 inflammasome is activated by double strand DNA (dsDNA) originating from bacteria, virus, and abnormal hosts [59,60,61,62,63]. Specific inflammasome complexes are defined by their subunits, and the precise conglomeration of various subunits to create these complexes depends on specific biologic activators [64,65]. Since the sensor proteins in inflammasomes are activated by specific activators, the sensor proteins interact with ASC and caspase-1 [64,65]. This interaction promotes the ASC speck formation and the cleavage of caspase-1, which are required for inflammasome activation [64,65]. Finally, the cleaved caspase-1 p10 fragment induces maturation and secretion of pro-inflammatory cytokines IL-1β and IL-18 [64,65]. 

Recent studies suggest a role for NLRP3 inflammasome activation in lung inflammation and fibrosis [66,67,68,69]. Elevated inflammasome activation has been reported to be augmented in human IPF lungs. Specifically, IL-18 protein levels were highly expressed in alveolar macrophages and epithelial cells [70]. Lasithiotaki et al. showed that transcriptional levels of inflammasome components, NLRP3 and caspase-1, were elevated in IPF patients when compared to the controls [71]. In addition, NLRP3 inflammasome activation by ATP was significantly increased in alveolar macrophages from IPF patients when compared to the controls [71]. Other studies have shown that inflammasome activation is present in other fibrotic lung diseases such as systemic sclerosis [72]. Fibroblasts from systemic sclerosis showed increased expression of multiple genes that was associated with the inflammasome or downstream signaling molecules [72]. Several studies have demonstrated a role for the NLRP3 inflammasome and its regulated cytokines in lung fibrosis in experimental lung fibrosis [68,73,74]. Given the increased prevalence of IPF in older populations, Stout-Delgado et al. investigated the role of NLRP3 inflammasome activation in age-dependent pulmonary fibrosis. Aging was found to be associated with the increased production of mitochondrial reactive oxygen species, which led to NLRP3 inflammasome activation and IL-1β and IL-18 secretion [73,75]. The role of NLRP3 inflammasome has also been tested in other pulmonary fibrosis models including ventilator induced lung fibrosis, silica-induced fibrosis, and particular manner-induced fibrosis [68,74,76]. Although the NLRP3 inflammasome is mainly expressed in innate immune cells, there are several reports of inflammasome independent function of NLRP3 in murine fibrosis models. Lv et al. demonstrated that mechanical stretch promotes the activation of endothelial-mesenchymal transition (EMT) and lung fibrosis via NLRP3-mediated signaling cascade [74]. Tian et al. showed that the NLRP3 inflammasome is activated in alveolar epithelial cells and that the NLRP3 inflammasome may regulate EMT through TGF-β1 [77]. While the study of the NLRP3 inflammasome has been explored in various lung fibrosis models, the role of AIM2 inflammasome has not been fully investigated. Terlizzi et al. demonstrated that the activation of the AIM2 inflammasome is increased in peripheral blood mononuclear cells (PBMCs) from patients with IPF. The AIM2 inflammasome activation contributes to the production and release of pro-fibrotic mediators [78]. Recently, our findings demonstrated that the expression and activation of AIM2 inflammasome expression and activation is enhanced in a lung fibrosis exacerbation model [79]. Therefore, recent studies suggest that the activation of NLRP3 or AIM2 inflammasome-dependent inflammation has a critical role in the progression of lung fibrosis.

### 3.2. The Mechanism of Inflammasome Activation in IPF

Several studies have shown the upstream mechanism by which the NLRP3 inflammasome is activated in animal fibrosis models. Vimentin, the most abundant intermediate filament in the cytoplasm, regulates the activation of NLRP3 inflammasome in the asbestos-induced lung injury and fibrosis model [80]. Inflammation, endothelial, and alveolar epithelial barrier permeability and fibrosis are attenuated in vimentin deficient mice after asbestos challenge, and vimentin-knockdown macrophages showed decreased active caspase-1 and IL-1β levels. Furthermore, in this study, the direct protein–protein interaction between NLRP3 and vimentin was demonstrated. Doster et al. showed that NLRP3 inflammasome activation is triggered by reactive oxygen species (ROS), which are generated by nicotinamide adenine dinucleotide phosphate (NADPH) oxidase (NOX) upon particle phagocytosis [81]. Inhibition of NADPH oxidase suppressed NLRP3 inflammasome activation and NLRP3 deficient mice had reduced inflammation and fibrosis after asbestos exposure. ROS can be repressed by autophagy [82]. Meng et al. demonstrated that the activation of autophagy suppressed angiotensin II-induced pulmonary fibrosis via the inhibition of ROS-derived NLRP3 inflammasome activation. Another study reported similar findings that autophagic dysfunction in alveolar epithelial cells leads to subsequent lung fibrosis in a silica nanoparticle-induced fibrosis model [83]. Metabolic stimulation can also stimulate NLRP3 inflammasome activation in lung fibrosis models. Xu et al. described that statin use is associated with interstitial lung abnormalities among smokers in chronic obstructive pulmonary disease (COPD) Gene, a well-defined large cohort of smokers [84]. In addition, they reported that statin administration increases bleomycin-induced lung inflammation and fibrosis via the NLRP3 inflammasome activation in a mouse model. While the upstream regulation of NLRP3 inflammasome activation has been well described in lung fibrosis models, the mechanism by which AIM2 inflammasome is activated in lung fibrosis has not been fully elucidated. Recently, we have described that glucose transport 1 (GLUT1)-dependent glycolysis regulates the exacerbation of lung fibrosis via AIM2 inflammasome activation [79]. We reported that GLUT1 deficiency ameliorates *S. pneumoniae*-mediated exacerbation of lung fibrosis in a mouse model [79]. In this study, we found that GLUT1 contributes to AIM2 inflammasome activation in macrophages [79]. We also found that the deficiency of GLUT1 suppressed the activation of AIM2 inflammasome in lung tissues under *S. pneumoniae*-mediated exacerbation of lung fibrosis [79].

## 4. Drosha ribonuclease III (DROSHA)-Dependent AIM2 Inflammasome Activation in IPF

### 4.1. The Activation of DROSHA in Alveolar Macrophages during IPF

While the changes of selective miRNA, which regulate differentiation, proliferation, and interaction between cell and cell, were identified and investigated in IPF, the role of DROSHA in alveolar macrophages, which are a critical cell type for pulmonary inflammation in IPF pathogenesis, remains unclear. The amount of evidence supports that macrophages play critical roles that affect fibrotic responses [85]. Alveolar macrophages produce transforming growth factor-beta 1 (TGF-β1), which promotes collagen accumulation, in both humans and mice [86,87]. These results suggest that a potential mechanism by which alveolar macrophages contribute to IPF. In addition to their role in TGF-β1 production, macrophages lead to the secretion of various inflammatory cytokines including TNF-α, IL-1, IL-6, IL-8, IL-10, and IL-12 [88]. Additionally, macrophages produce various chemokines such as CXCL1, CXCL2, CXCL9, CXCL10, CXCL12, CCL5, CCL17, and CCL18 [88]. Additionally, the macrophage-derived production of lipid mediators including eicosanoids might contribute to fibrosis [89]. On the other hand, the function of lipid mediators has not been fully identified, further investigation is needed in patients with IPF and a mouse model of lung fibrosis. Macrophages also regulate ECM remodeling by the secretion of matrix metalloproteinases and by the rearrangement of collagen [90,91]. From a different aspect, macrophages could determine the metabolic fate of environmental cells, which might conduct the glycolytic reprogramming in fibroblast during fibrosis [92,93]. Additionally, macrophages produce vascular endothelial growth factor (VEGF), which is related to both pro-fibrotic responses or anti-fibrotic responses [94,95,96]. It depends on the pattern of expression of angiogenic factors and the target cell. Therefore, macrophages are critical for the development of pulmonary fibrosis.

In other cells, the function of DROSHA has been reported [17,21,97]. In smooth muscle cells, the roles of DROSHA are required for cell survival [97]. Additionally, DROSHA regulates stress-induced death by phosphorylation of its N terminus via p38 MAPK [17]. The translocation of DROSHA by phosphorylation into cytosol from the nucleus is the regulatory mechanism of stress-induced death [21]. In the function of T cells, DROSHA is critical for T cell compartment. Deficiency of DROSHA results in T lymphopenia, particularly in the CD8+ compartment [21]. This study suggests that DROSHA-dependent miRNA production is required for the homeostasis of mature T cells [86]. In our recent study, we showed an increase in the DROSHA protein expression levels in alveolar macrophages during IPF [10]. Immunohistochemistry staining indicated that the protein levels of DROSHA were significantly increased in lung tissues from patients with IPF compared to non-IPF patients [10]. In particular, the intensity and number of DROSHA-positive staining in CD68-positive alveolar macrophages were significantly increased in lung tissues from patients with IPF [10]. These results showed that the high levels of DROSHA in alveolar macrophages contributed to pulmonary inflammation in patients with IPF [10]. Similarly, the protein levels of DROSHA were elevated in alveolar macrophages in a mouse model of bleomycin-induced pulmonary fibrosis [10]. In other words, the expression levels of DGCR8 were not changed in lung tissues between patients with IPF and non-IPF patients [10]. Our results demonstrated that the elevation of DROSHA in alveolar macrophages is critical for pulmonary inflammation in IPF pathogenesis [10]. 

### 4.2. The Role of DROSHA in AIM2 Inflammasome Activation

Among the various inflammasomes, the mechanism of AIM2 inflammasome activation remains unclear. Currently, the representative mechanism of AIM2 inflammasome activation has been identified by the recognition of dsDNA by AIM2 [59,60,61,62,63] or the increase of AIM2 expression levels [98]. These two mechanisms are required for the AIM2 inflammasome complex formation, which causes the activation of caspase-1 p10 via cleavage of pro-caspase-1. 

In our recent study, we suggest that DROSHA is an upstream regulator of AIM2 inflammasome activation [10]. We showed that the deficiency of DROSHA suppresses the activation of caspase-1 by AIM2 inflammasome activation in alveolar macrophages [10]. The deficiency of DROSHA reduces the apoptosis-associated speck-like protein aontaining a CARD (ASC) speck formation in the AIM2 inflammasome complex formation during AIM2 inflammasome activation [10]. These results suggest that DROSHA is critical for the activation of caspase-1 and AIM2 inflammasome complex formation during AIM2 inflammasome activation [10]. As further investigation of the molecular mechanism for DROSHA-dependent AIM2 inflammasome activation is conducted, the understanding of the cellular pathway related to the regulation of DROSHA expression could be important for the discovery of a new mechanism for AIM2 inflammasome activation. Previous studies have shown that the expression of DROSHA is tightly controlled by various cellular pathways such as post-translational modifications, protein degradation pathways, and alternative splicing [88,89,90,91,92]. Link et al. showed that the alternative splicing of DROSHA causes the alteration of the subcellular localization of DROSHA between the cytoplasm and nucleus [14,99]. They suggest that the cytosolic DROSHA potentially alters the function depending on the interaction with partners such as substrate RNAs or regulatory mechanisms [100,101]. Additionally, Ye et al. showed that E3 ubiquitin-protein ligase Mdm2 directly interacts with DROSHA [102]. Its interaction induces the ubiquitination of DROSHA, which causes the degradation of DROSHA protein [103]. Therefore, further investigation for the molecular mechanisms that affect DROSHA expression via alternative splicing or ubiquitination of DROSHA under IPF pathogenesis in AIM2 inflammasome activation is needed. 

### 4.3. miRNA as DAMPs in AIM2 Inflammasome Activation

AIM2 inflammasome is activated by the recognition of cytosolic self-dsDNA as well as foreign dsDNA from pathogens [99]. AIM2 has been shown to detect dsDNA in a sequence-independent manner [62,63]. Since miRNAs, which are produced by DROSHA, have a hairpin structure with double-stranded RNA (dsRNA), miRNAs could make hetero-triplex structures in the location of specific sequences of DNA through the interaction with dsDNA [13]. Recent reports suggest that sub-cellular localization is critical to miRNA function [104]. Among the miRNA compartment, the localization of miRNA in the cytoplasm is important for RNA granules, endomembranes, and the export of miRNA to extracellular space [104]. Since cytosolic miRNAs have double-strand structure as well as dsDNA, cytosolic miRNA could be an activator of the AIM2 inflammasome. In our recent study, we found that the transduction of miRNAs in a sequence-independent manner promoted the activation of the AIM2 inflammasome and the cleavage of caspase-1 in macrophages [10]. Moreover, the transduction of miRNAs increased the maturation and secretion of IL-1β and IL-18 [10]. Furthermore, transduction of miRNAs increased the protein complex formation of AIM2 inflammasome by ASC speck formation, which is important for AIM2 inflammasome activation [10]. These results suggest that the double-stranded structure of cytosolic miRNAs induces the AIM2 inflammasome activation via AIM2 inflammasome complex formation (Figure 3). 

DAMPs could contribute to lung inflammation and fibrosis during IPF. There are various types of DAMPs that come from different origins. The simple type of DAMPs might be intracellular components including nucleic acids such as DNA or RNA and organelles that are released from dying and injured cells. DAMPs might be released by cells through the transportation via the processes using endosomes and membrane bound vesicles. Among the various types of DAMPs, certain DAMP can be produced by the characteristic alteration of intracellular proteins such as the transformation from collagens to collagen fragments. These kinds of DAMPs are recognized by innate immune receptors and the activated pro-inflammatory signaling pathway. The activation of these immune receptors could be protective or harmful events, which is determined by the type of DAMP and DAMP-related specific receptor. In IPF, the role of DAMPs has been investigated in IL-17A receptor (IL-17RA)-mediated NF-kB signaling and the role of high mobility group box 1 (HMGB1) during inflammation and fibrosis [102,105]. IL-17A stimulates proliferation and survival of airway smooth muscle cells via the activation of IL-17RA [106]. The stimulation of IL-17RA leads to changes to various cellular pathways including cell proliferation, myofibroblast differentiation, and the production of extracellular matrix (ECM) proteins [102]. The levels of HMGB1 in BAL were increased in patients with IPF compared the control [105]. In bleomycin-induced pulmonary fibrosis in mice, HMGB1 protein levels were elevated in bronchiolar epithelial cells at early stage and in alveolar epithelial and inflammatory cells at late stage [105]. Recent study showed that the levels of miRNAs were increased in bronchoalveolar lavage (BAL) from patients with IPF relative to the controls [104]. These findings showed that the miRNA in BAL could contribute to damage-associated molecular pattern (DAMP)-dependent chronic lung inflammation in the progression of IPF [104,105]. In our recent study, we showed that the intracellular excessive accumulation of miRNA promotes AIM2 inflammasome-dependent caspase-1 activation in alveolar macrophages [10]. Moreover, the intracellular excessive accumulation of miRNA increases the ASC speck formation that is required for AIM2 inflammasome complex formation. Our results suggest that miRNA could be a critical DAMP for AIM2 inflammasome activation in pulmonary inflammation during IPF [10]. Based on our observation, the high levels of miRNA in BAL or blood in patients with chronic lung diseases might be a prognostic biomarker for the prediction of IPF. 

### 4.4. Therapeutic Approach of AIM2 Inflammasome Activation

Recent studies have shown that potential inhibitors have been identified as the NLRP3 inflammasome inhibitor [107,108,109,110,111,112]. The effects of potential NLRP3 inflammasome inhibitors such as OLT1177, Tranilast, Oridonin, CY-09, and MCC950 were demonstrated in cells and a representative mouse model for high fat diet (HFD)-induced diabetes, acute arthritis, and Parkinson’s disease [107,108,109,110,111,112]. Each potential NLRP3 inflammasome inhibitor reduced the activation of NLRP3 inflammasome through the alteration of upstream pathways for NLRP3 inflammasome activation including mitochondrial dysfunction and potassium efflux [107,108,109,110,111,112]. Unlike the NLRP3 inflammasome, the understanding of potential inhibitor for AIM2 inflammasome activation is still unclear. Recent studies showed that the potential molecules that could inhibit AIM2 including pyrin-only proteins (POPs), CARD-only proteins (COPs), and TRIM11 [113,114]. In particular, pyrin-only proteins 1 (POP1) and POP3 suppressed the activation of AIM2 inflammasome via the interference of PYD-PYD interaction [115,116]. Additionally, IFI16-β, as a new inhibitor of AIM2 inflammasome activation, was investigated for new knowledge for the design and development of anti-inflammatory agents [117]. In human chronic skin disease, cathelicidin peptide LL-37 inhibits the activation of the AIM2 inflammasome by interfering cytosolic DNA in keratinocytes [118]. This result suggests that cathelicidin LL-37 might be a therapeutic target of the AIM2 inflammasome to ameliorate cutaneous inflammation in chronic skin disease. 

In cancer, increased expression of AIM2 has been reported in nasopharyngeal carcinoma tumors [119,120], oral squamous cell carcinoma [121], and lung adenocarcinoma [122]. On the other hand, AIM2 has been shown to suppress the development of cancer such as colorectal cancer [123,124]. Reduced expression and instability of the AIM2 have been reported in cancer tissues from patients with colorectal cancer [124,125,126]. The reduction of AIM2 expression is linked to a poorer prognosis in colorectal cancer patients [127]. Furthermore, the reduction of AIM2 expression was found in prostate cancer [128]. In different types of tumor tissue, the variation of AIM2 expression may play a role in the pathogenesis of various cancers. Since the role and activation of AIM2 are different in each type of cancer, the drug discovery for the therapeutic target of the AIM2 inflammasome should be conducted carefully. 

Currently, the selective chemical inhibitor for the AIM2 inflammasome has not been identified as a therapeutic agent in human diseases such as IPF. In future investigations, the finding and validation of the selective chemical inhibitor for the AIM2 inflammasome might be important for the treatment of IPF.

## 5. Conclusions

In this review, the understanding of the DROSHA–miRNA–AIM2 inflammasome axis in the pathogenesis of IPF could be helpful to further investigate the field of interaction between the miRNA and AIM2 inflammasome during IPF. The evidence for the importance of AIM2 inflammasome activation in IPF and other lung diseases could provide the necessity of developing therapeutic agents for the AIM2 inflammasome [129,130,131]. Recent reports imply that the activation of the AIM2 inflammasome or other inflammasomes is a critical event during the severity of asthma and chronic obstructive pulmonary disease (COPD) [129,130,131]. A new study, which could discover a therapeutic approach to the control of inflammasomes in various lung diseases, is needed. Furthermore, the novel strategy for the inhibition of DROSHA such as specific chemical inhibitors or therapeutic agents on human clinical trials is still unclear. As a therapeutic approach for human diseases including IPF, a high throughput platform to discover and validate specific DROSHA inhibitors should be started in the near future. Therefore, the novel approach for the development of the therapeutic agents belonging to the AIM2 inflammasome should be attempted by global scientists who have the passion to cure human lung diseases. 

## Figures and Tables

**Figure 1 ijms-21-01668-f001:**
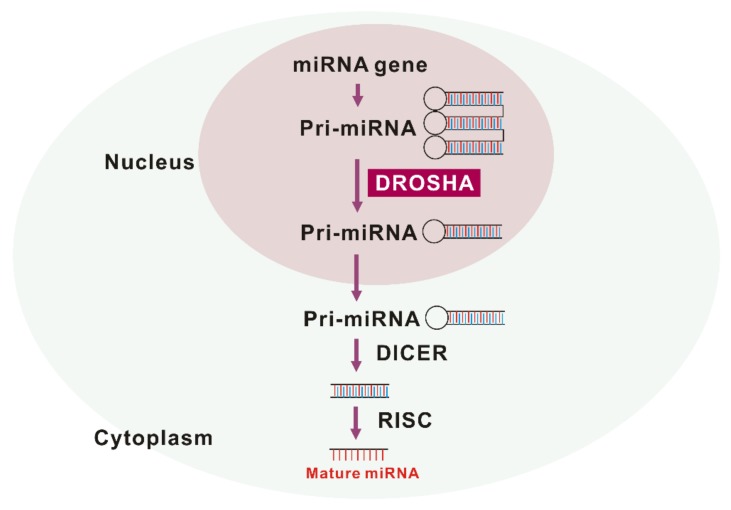
The maturation of miRNA by drosha ribonuclease III (DROSHA).

**Figure 2 ijms-21-01668-f002:**
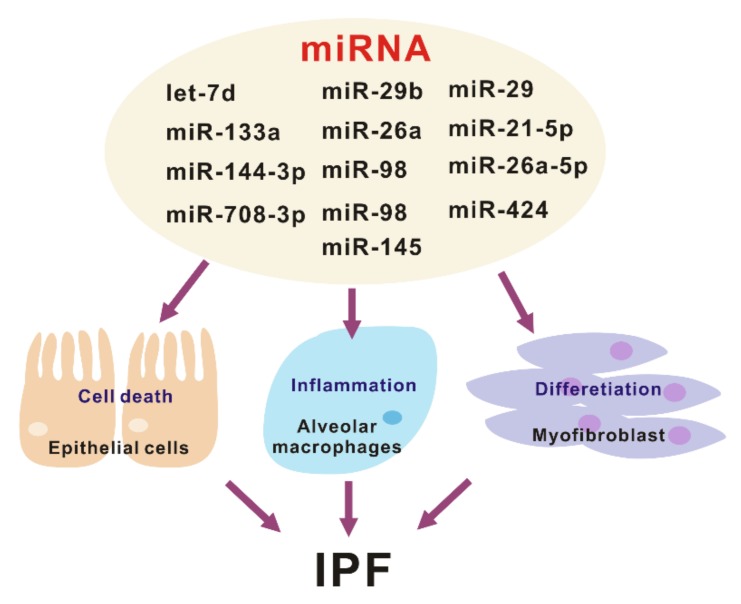
The various roles of miRNA in the pathogenesis of idiopathic pulmonary fibrosis (IPF).

**Figure 3 ijms-21-01668-f003:**
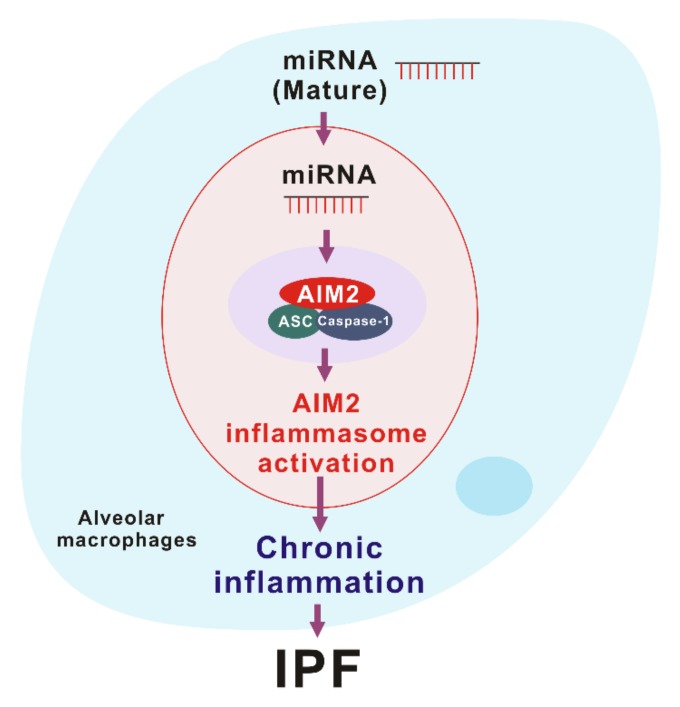
The role of miRNA-dependent AIM2 inflammasome activation in idiopathic pulmonary fibrosis (IPF).

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
