# Peer review of "DROSHA-Dependent miRNA and AIM2 Inflammasome Activation in Idiopathic Pulmonary Fibrosis"

_ijms, 2020, doi:10.3390/ijms21051668_

Round 1

Reviewer 1 Report

This review focuses in AIM2 inflammasome activation in idiopathic pulmonary fibrosis and is generally well written, although improvements should be made to improve clarity and readability.

Comments:

The text should be edited throughout the manuscript to improve sentence construction and similar concerns. Abstract, sentence 1, change idiopathic interstitial fibrosis to idiopathic pulmonary fibrosis. 2.1 - include a schematic on miRNA processing highlighting role of Drosha. 2.2 - Sentence 1, revise to indicate that Drosha is involved in maturation or processing and not biogenesis of miRNAs Line 91, change medicated to mediated. Line 103, Change PFAL to PFRL Fig. 1 is not very informative. Can you improve it ? Remove the background lung drawing. It appears as if one lung produces miRNAs to affect the other lung! Line 207, change elevation to activation. Line 247, change inflammasome to inflammasomes. Lines 301 -317, references are missing. Fig. 2 is not very informative. Can you improve it? Remove the background lung drawing.  5. Conclusion - Conclusion section is very brief and abrupt. It should highlight important findings reviewed and provide future perspectives. Important references pertaining to the subject matter reviewed appear to have been omitted, examples include Colarusso et al 2019, Front. Pharmacy; Sorrentino et al 2017, Our Respiratory J; Santos et al 2012, AJP Lung Cell Mol Physiol.

Author Response

Response to IJMS Reviewer 1 Comments

The text should be edited throughout the manuscript to improve sentence construction and similar concerns.

Reviewer’s Comment 1

Abstract, sentence 1, change idiopathic interstitial fibrosis to idiopathic pulmonary fibrosis.

Response 1

As reviewer’s comment, we changed as idiopathic pulmonary fibrosis.

The following text has been added to Page 1. Line 17:

Page 1. Line 17 “Idiopathic pulmonary fibrosis (IPF) is a chronic, progressive interstitial lung disease.”

Reviewer’s Comment 2

2.1 - include a schematic on miRNA processing highlighting role of Drosha.

Response 2

As reviewer’s comment, we provided a schematic on miRNA processing by DROSHA.

Please see new Figure 1.

Reviewer’s Comment 3

2.2 - Sentence 1, revise to indicate that Drosha is involved in maturation or processing and not biogenesis of miRNAs

Response 3

We agree with your opinion. As reviewer’s comment, we revised the explanation of DROSHA.

The following text has been added to Page 2. Line 78:

Page 2. Line 78 “DROSHA is involved in maturation or processing and not biogenesis of microRNAs (miRNAs) which are short non-coding RNA molecules composed of 18-25 nucleotides.”

Reviewer’s Comment 4

Line 91, change medicated to mediated.

Response 4

As reviewer’s comment, we changed as mediated.

The following text has been added to Page 2. Line 90:

Page 2. Line 90 “estrogen receptor (ER)-mediated pulmonary fibrosis and this study suggested that let-7 dependent ER.”

Reviewer’s Comment 5

Line 103, Change PFAL to PFRL

Response 5

As reviewer’s comment, we changed as PFRL.

The following text has been added to Page 3. Line 103:

Page 3. Line 103 “Mechanistically, PFRL promoted lung fibroblast activation by acting as a competing endogenous”

Reviewer’s Comment 6

Fig. 1 is not very informative. Can you improve it ? Remove the background lung drawing. It appears as if one lung produces miRNAs to affect the other lung!

Response 6

As reviewer’s comment, we provided a schematic with more information for the role of miRNAs in IPF.

Please see new Figure 2.

Reviewer’s Comment 7

Line 207, change elevation to activation.

Response 7

As reviewer’s comment, we changed as activation.

The following text has been added to Page 5. Line 207:

Page 5. Line 207 “The activation of DROSHA in alveolar macrophages during IPF.”

Reviewer’s Comment 8

Line 247, change inflammasome to inflammasomes.

Response 8

As reviewer’s comment, we changed as inflammasomes.

The following text has been added to Page 6. Line 248:

Page 6. Line 248 “Among the various inflammasomes, the mechanism of AIM2 inflammasome activation remains”

Reviewer’s Comment 9

Lines 301 -317, references are missing.

Response 9

As reviewer’s comment, we provided references for previous other study and our study in Line 301-317.

The following text has been added to Page 7 and 8. Line 303-317:

Page 7 and 8. Line 303-317 “The levels of HMGB1 in BAL were increased in patients with IPF compared control [107]. In bleomycin-induced pulmonary fibrosis in mice, HMGB1 protein levels were elevated in bronchiolar epithelial cells at early stage and in alveolar epithelial and inflammatory cells at late stage [107]. Recent study showed that the total amount of miRNA was elevated in bronchoalveolar lavage (BAL) from patients with IPF compared to controls [105]. These previous studies indicated that the miRNA in BAL could have a role in damage-associated molecular pattern (DAMP)-dependent chronic lung inflammation in the progression of IPF [105,107]. In recent our study, we showed that the intracellular excessive accumulation of miRNA promotes AIM2 inflammasome-dependent caspase-1 activation in alveolar macrophages [10]. Moreover, the intracellular excessive accumulation of miRNA increases the ASC speck formation which is required for AIM2 inflammasome complex formation. Our results suggest that miRNA could be a critical DAMP for AIM2 inflammasome activation in pulmonary inflammation during IPF [10]. Based on our observation, the high levels of miRNA in BAL or blood in patients with chronic lung diseases might be a prognostic biomarker for prediction of IPF.”

Reviewer’s Comment 10

Fig. 2 is not very informative. Can you improve it? Remove the background lung drawing.

Response 10

As reviewer’s comment, we provided a schematic with more information for the role of miRNA-dependent AIM2 inflammasome activation in IPF.

Please see new Figure 3.

Reviewer’s Comment 11

  1. Conclusion - Conclusion section is very brief and abrupt. It should highlight important findings reviewed and provide future perspectives. Important references pertaining to the subject matter reviewed appear to have been omitted, examples include Colarusso et al 2019, Front. Pharmacy; Sorrentino et al 2017, Our Respiratory J; Santos et al 2012, AJP Lung Cell Mol Physiol.

Response 11

As reviewer’s comment, we provided the additional description for significance in the lung field and potential future of current review in conclusion.

The following text has been added to Page 8. Line 351:

Page 8. Line 351 “The evidences for the importance of AIM2 inflammasome activation in IPF and other lung diseases could provide the necessity of development of therapeutic agents for AIM2 inflammasome [131-133]. Recent reports implicate the activation of AIM2 inflammasome or other inflammasomes is a critical event during the severity of asthma and chronic obstructive pulmonary disease (COPD) [131-133]. The new study which could discover therapeutic approach to control of inflammasomes in various lung diseases is needed. Also, the novel strategy for the inhibition of DROSHA such as specific chemical inhibitor or therapeutic agents on human clinical trials is still unclear. As a therapeutic approach for human diseases including IPF, the high throughput platform to discover and validate specific DROSHA inhibitors should be start in near future. Therefore, the novel approach for the development of the therapeutic agents belong to AIM2 inflammasome should be try by the global scientists who have the passion to cure human lung diseases.”

Reviewer 2 Report

In this review the authors provide a very comprehensive overview of the role miRNA in general, and of DORSHA-dependent miRNA and inflammasome activation in IPF specifically. It is a very complete overview and as such in parts perhaps a little too dense in the listing of the impressive amount of knowledge in this field, which combined with a variable number of syntax and grammar errors makes for slightly cumbersome reading in places. Nonetheless, it is an impressive and state-of-the-art review.

Specific comments:

  1. p.1 l.30: 'idiopathic interstitial fibrosis' should read 'idiopathic pulmonary fibrosis' 
  2. p.1 l.32: 'populations' should be changed to 'inhabitants'
  3. p.1 l.32: 'medial' survival should probably read 'median' survival
  4. p.1 l.41: 'parastitial' fibrosis does not exist; do the authors mean 'paraseptal'?
  5. p.4 l.138: do the authors mean viral particles?
  6. p.4 l.140: what is meant by 'aberrant' host? Please rephrase to clarify
  7. p. 5 l.218: 'The other hands'... Should this be 'on the other hand'?
  8. The conclusion is too short and could be expanded to clarify the significance for the field and potential future lines of investigation / treatment

Author Response

Response to IJMS Reviewer 2 Comments

In this review the authors provide a very comprehensive overview of the role miRNA in general, and of DORSHA-dependent miRNA and inflammasome activation in IPF specifically. It is a very complete overview and as such in parts perhaps a little too dense in the listing of the impressive amount of knowledge in this field, which combined with a variable number of syntax and grammar errors makes for slightly cumbersome reading in places. Nonetheless, it is an impressive and state-of-the-art review.

Specific comments:

Reviewer’s Comment 1

p.1 l.30: 'idiopathic interstitial fibrosis' should read 'idiopathic pulmonary fibrosis'

Response 1

As reviewer’s comment, we changed as idiopathic pulmonary fibrosis.

The following text has been added to Page 1. Line 30.

Page 1. Line 30 “Idiopathic pulmonary fibrosis (IPF) is a chronic, progressive and lethal disorder of unknown”

Reviewer’s Comment 2

p.1 l.32: 'populations' should be changed to 'inhabitants'

Response 2

As reviewer’s comment, we changed as inhabitants.

The following text has been added to Page 1. Line 32.

Page 1. Line 32 “America estimated between 6 and 18 per 100,000 inhabitants [1].”

Reviewer’s Comment 3

p.1 l.32: 'medial' survival should probably read 'median' survival

Response 3

As reviewer’s comment, we changed as median.

The following text has been added to Page 1. Line 32.

Page 1. Line 32 “The median survival period of IPF”

Reviewer’s Comment 4

p.1 l.41: 'parastitial' fibrosis does not exist; do the authors mean 'paraseptal'?

Response 4

As reviewer’s comment, we changed as paraseptal.

The following text has been added to Page 1. Line 41.

Page 1. Line 41 “heterogeneity with areas of subpleural and paraseptal fibrosis and honeycombing [6].”

Reviewer’s Comment 5

p.4 l.138: do the authors mean viral particles?

Response 5

As reviewer’s comment, we changed as viral particles.

The following text has been added to Page 4. Line 138.

Page 4. Line 138 “inflammasome assembly and activation are mediated by various bacteria and viral particles whereas”

Reviewer’s Comment 6

p.4 l.140: what is meant by 'aberrant' host? Please rephrase to clarify

Response 6

As reviewer’s comment, we changed as abnormal. Abnormal is a better expression than aberrant.

The following text has been added to Page 4. Line 141.

Page 4. Line 141 “and abnormal hosts [59-63]. Specific inflammasome complexes are defined by their subunits, and the”

Reviewer’s Comment 7

  1. 5 l.218: 'The other hands'... Should this be 'on the other hand'?

Response 7

As reviewer’s comment, we changed as on the other hand.

The following text has been added to Page 5. Line 219.

Page 5. Line 219 “On the other hand, the function of lipid mediators has”

Reviewer’s Comment 8

The conclusion is too short and could be expanded to clarify the significance for the field and potential future lines of investigation / treatment

Response 8

As reviewer’s comment, we provided the expanded conclusion section for the field and potential future lines of investigation / treatment in conclusion section.

The following text has been added to Page 8. Line 353-364.

Page 8. Line 351-364 “The evidences for the importance of AIM2 inflammasome activation in IPF and other lung diseases could provide the necessity of development of therapeutic agents for AIM2 inflammasome [131-133]. Recent reports implicate the activation of AIM2 inflammasome or other inflammasomes is a critical event during the severity of asthma and chronic obstructive pulmonary disease (COPD) [131-133]. The new study which could discover therapeutic approach to control of inflammasomes in various lung diseases is needed. Also, the novel strategy for the inhibition of DROSHA such as specific chemical inhibitor or therapeutic agents on human clinical trials is still unclear. As a therapeutic approach for human diseases including IPF, the high throughput platform to discover and validate specific DROSHA inhibitors should be start in near future. Therefore, the novel approach for the development of the therapeutic agents belong to AIM2 inflammasome should be try by the global scientists who have the passion to cure human lung diseases.”